# Melatonin in Glaucoma: Integrative Mechanisms of Intraocular Pressure Control and Neuroprotection

**DOI:** 10.3390/biomedicines13051213

**Published:** 2025-05-16

**Authors:** Xinyu Hou, Yingzi Pan

**Affiliations:** Department of Ophthalmology, Peking University First Hospital, No. 8 Xi Shi Ku Street, Xi Cheng District, Beijing 100034, China; 2311210252@stu.pku.edu.cn

**Keywords:** melatonin, glaucoma, intraocular pressure, retinal ganglion cells, neuroprotection, nanocarriers, oxidative stress, mitochondrial dysfunction, circadian regulation

## Abstract

**Background**: Glaucoma is a leading cause of irreversible visual loss worldwide, characterized by progressive retinal ganglion cell (RGC) degeneration and optic nerve damage. Current therapies mainly focus on lowering intraocular pressure (IOP), yet fail to address pressure-independent neurodegenerative mechanisms. Melatonin, an endogenously produced indoleamine, has gained attention for its potential in modulating both IOP and neurodegeneration through diverse cellular pathways. This review evaluates the therapeutic relevance of melatonin in glaucoma by examining its mechanistic actions and emerging delivery approaches. **Methods**: A comprehensive literature search was conducted via PubMed and Medline to identify studies published between 2000 and 2025 on melatonin’s roles in glaucoma. Included articles discussed its effects on IOP regulation, RGC survival, oxidative stress, mitochondrial integrity, and inflammation. **Results**: Evidence supports melatonin’s involvement in IOP reduction via MT receptor activation and its synergism with adrenergic and enzymatic regulators. Moreover, it protects RGCs by mitigating oxidative stress, preventing mitochondrial dysfunction, and inhibiting apoptotic and inflammatory cascades. Recent advances in ocular drug delivery systems enhance its bioavailability and therapeutic potential. **Conclusions**: Melatonin represents a multi-target candidate for glaucoma treatment. Further clinical studies are necessary to establish optimal dosing strategies, delivery methods, and long-term safety in patients.

## 1. Introduction

Glaucoma is a group of optic nerve neuropathies characterized mainly by progressive degeneration of RGC and optic nerve injuries. Elevated IOP plays an important role in the development of glaucoma [1,2]. In 2020, an estimated 76 million people worldwide were affected by glaucoma, of which approximately 74% had primary open-angle glaucoma and the remainder comprised mainly angle-closure and normal-tension subtypes [3]. Glaucoma accounts for roughly 3.6 million cases of bilateral blindness globally, and no current therapy can restore lost vision, underscoring the imperative for early detection and management. With global prevalence projected to rise to 112 million by 2040 due to population aging [4], optimized screening and innovative treatments are urgently needed. As aging populations contribute to a significant increase in glaucoma prevalence in the coming decades [4], improved detection methods and enhanced treatment strategies will be essential to prevent avoidable blindness.

While the cellular mechanisms underlying glaucomatous damage remain inadequately understood, it is widely recognized that the primary injury occurs at the optic nerve head and may involve multiple processes [5], including ischemia [6], axoplasmic transport obstruction [7], biomechanical stress on axons [8], and secondary effects from RGC axon loss [9]. Additionally, optic nerve head glial remodeling in glaucoma involves the activation of resident astrocytes and an extracellular matrix [10]. These glial cells within the optic nerve head and retina respond to mechanical and metabolic stress and strain, playing a critical role in disease progression [9,11,12]. These converging factors underscore glaucoma’s multifactorial pathology.

Melatonin, the principal hormone of the pineal gland, has emerged as a promising neuroprotective agent in this context. Studies involving adults with normal vision indicate that the system responsible for detecting brightness provides direct photic input to the suprachiasmatic nuclei (SCN), the primary circadian pacemaker in mammals, through neural pathways independent of standard retinal projections associated with vision [13,14]. Two distinct mechanisms facilitate photic entrainment (Figure 1).The primary mechanism, the retinohypothalamic tract (RHT), channels light-derived signals to the SCN and pineal gland. The RHT transmits light signals from melanopsin-expressing intrinsically photosensitive retinal ganglion cells (ipRGCs) to the SCN through glutamate and pituitary adenylate cyclase-activating polypeptide (PACAP) [15], directly resetting circadian clocks via NMDA/PAC1 receptor activation [16,17]. The secondary pathway, termed the geniculohypothalamic tract (GHT), originating in the thalamic intergeniculate leaflet (IGL), relays non-photic behavioral and metabolic inputs via neuropeptide Y (NPY) and GABA to modulate SCN phase plasticity [18,19].

Downstream, SCN efferents project to the paraventricular nucleus (PVN), which relays signals via the spinal intermediolateral column (IML) and superior cervical ganglion (SCG) to regulate pineal melatonin secretion, suppressing melatonin synthesis in light while allowing nocturnal melatonin release [20].

As retinal ganglion cells with photoreceptive functions play a crucial role in synchronizing the SCN, and ocular disorders can disrupt circadian rhythms, the potential link between melatonin and visual health has become a focal point of research [21,22].

## 2. Materials and Methods

A comprehensive literature search was conducted using the PubMed and Medline databases to identify relevant publications up to 30 March 2025. The search focused on articles published between 2000 and 2025, employing the following keywords and combinations: “melatonin”, “glaucoma”, “*N*-acetyl-5-methoxytryptamine”, “RGC protection”, “intraocular pressure”, and “neuroprotection”.

Studies not published in English, those irrelevant to the scope of this review, or those lacking complete research data were excluded. Articles identified as errata or corrections were also removed. The literature screening and selection process was independently conducted by two authors. Discrepancies in article inclusion were resolved through open discussion with senior co-authors to ensure consensus.

Interventionary studies involving animals or humans and other studies requiring ethical approval must list the authority that provided approval and the corresponding ethical approval code.

## 3. Melatonin Synthesis and Physiological Functions

Melatonin (*N*-acetyl-5-methoxytryptamine), an indolamine hormone derived from tryptophan, is synthesized primarily in pinealocytes and functions as an endocrine modulator of circadian rhythms across multiple organ systems [23]. Common dietary sources of melatonin include walnuts, tart cherries, grapes, tomatoes, rice, and oats; consumption of these foods has been shown to modestly elevate systemic—and by extension ocular—melatonin levels [24]. Melatonin’s ocular effects are closely intertwined with other neurotransmitters such as dopamine; in the retina, melatonin and dopamine form an antagonistic circadian feedback loop, with melatonin predominating at night and dopamine during the day. Disruption of this balance in glaucoma may contribute to the circadian rhythm disturbances observed in patients. In addition to the pineal gland, retinal photoreceptors have been identified as local production sites for melatonin [25,26], where it seemingly acts in a paracrine manner due to its minimal systemic release. Further studies have indicated its biosynthesis in the ciliary body epithelium, with subsequent secretion into the aqueous humor [27,28,29].

Melatonin biosynthesis proceeds via four distinct enzymatic reactions (Figure 2). The metabolic sequence begins with hydroxylation of tryptophan by tryptophan hydroxylase (TPH), generating 5-hydroxytryptophan. This intermediate is then transformed into serotonin through a decarboxylation step mediated by aromatic amino acid decarboxylase (AADC) [30]. Subsequently, arylalkylamine *N*-acetyltransferase (AANAT) catalyzes the acetylation of serotonin into *N*-acetylserotonin [31], which is ultimately methylated by hydroxyindole-*O*-methyltransferase (HIOMT) to form melatonin [32]. This biosynthetic pathway is subject to circadian control, with both enzymatic activity and gene transcription of TPH and AANAT exhibiting time-of-day dependency, especially nocturnal upregulation [15,33]. A comparable daily rhythm of melatonin synthesis has been identified in the retina, reinforcing its role in local circadian regulation [34,35]. Once synthesized, melatonin interacts with a diverse array of receptor subtypes located across multiple tissues. These interactions mediate its systemic and local functions, including regulation of sleep architecture, endocrine cycles, reproductive timing, immune modulation, and behavioral rhythms.

### 3.1. Regulation of Intraocular Pressure by Melatonin

Circadian variation in melatonin concentration within the aqueous humor has been well documented, with peak levels aligning with the nighttime phase, corresponding to fluctuations observed in systemic circulation [36]. The discovery that melatonin is locally synthesized in the ciliary body, along with the localization of its receptors on non-pigmented epithelial cell membranes [28,29,37], suggests an intrinsic role for melatonin in modulating intraocular dynamics, including aqueous fluid secretion and pressure regulation. Because these concentrations rise in the dark phase—preceding the typical early morning decline in IOP [27,38]—a regulatory link between melatonin and IOP has been proposed, as summarized in Appendix A.

Research by Alcantara-Contreras and colleagues demonstrated that MT1 receptor-deficient mice exhibited elevated nighttime IOP and significant RGC loss, supporting a possible association between impaired melatonin signaling and glaucoma susceptibility [39,40]. In parallel, a decline in receptor expression during glaucoma progression, as described by Martínez-Águila et al., raises concerns about therapeutic efficacy depending on residual receptor availability [41]. Numerous structural analogs of melatonin—including 5-MCA-NAT and other derivatives such as 6-chloromelatonin and 2-iodomelatonin—have shown promising hypotensive activity in ocular models [42].

Melatonin interacts with three receptor subtypes: MT1, MT2, and the putative MT3 receptor. Several studies have demonstrated that MT3 activation plays a crucial role in IOP reduction. Pintor et al. first reported that the MT3-selective agonist 5-MCA-NAT lowered IOP more effectively than melatonin in New Zealand white rabbits, with reductions of 43% and 24%, respectively [43]. Further research by Serle et al. in glaucomatous monkeys showed that 5-MCA-NAT reduced IOP progressively over five days, with effects lasting more than 18 h [44]. These findings indicate that MT3 receptors mediate a significant portion of melatonin’s ocular hypotensive effects.

Apart from MT3 activation, MT2 receptors have also been implicated in IOP regulation. Alarma-Estrany et al. demonstrated that MT3-selective analog INS48848 reduced IOP by 36% in normotensive rabbits. Beyond analogs, oral melatonin supplementation (3–10 mg nightly) has been shown in healthy volunteers to lower next-morning IOP by 1–2 mmHg over several days, suggesting a feasible dietary or nutraceutical approach [45]. Additionally, studies by Dortch-Carnes and Tosini demonstrated that melatonin, 5-MCA-NAT, and IIK7 (an MT2 agonist) reduced nitric oxide (NO) and cyclic GMP (cGMP) production in human ciliary epithelial cells, suggesting that MT2 activation plays a role in aqueous humor regulation [46]. Martínez-Águila et al. examined the effects of melatonin and 5-MCA-NAT on IOP in control and glaucomatous mice. Melatonin reduced IOP by 19.4% in controls and 32.6% in glaucomatous mice, while 5-MCA-NAT had similar effects and prevented IOP progression. Receptor antagonist studies confirmed MT2 involvement for melatonin and MT3 for 5-MCA-NAT. These findings suggest melatonin analogs as potential glaucoma treatments, with 5-MCA-NAT offering preventive benefits [47].

Melatonin’s hypotensive effects are further enhanced by its interactions with the adrenergic system. Crooke et al. found that 5-MCA-NAT modulated *β*2/*α* 2A-adrenergic receptor expression in rabbit ciliary epithelial cells, enhancing the IOP-lowering effects of timolol (by 14–16.75%) and brimonidine (by 29.26–39.07%) [48]. These results suggest that melatonin analogs may have synergistic effects when used in combination with conventional glaucoma medications.

The sympathetic nervous system has also been shown to modulate melatonin’s ocular hypotensive action. Alarma-Estrany et al. reported that the MT2 agonist IIK7 reduced IOP by 38.5% in rabbits, but chemical sympathectomy significantly diminished this effect [45]. Furthermore, *β*-adrenergic agonists enhanced IIK7’s IOP-lowering properties, highlighting the importance of sympathetic regulation in melatonin-mediated ocular hypotension.

Another mechanism through which melatonin regulates IOP is its effect on aqueous humor secretion. Li et al. demonstrated that melatonin increased chloride (Cl^−^) transport and Aqueous humor secretion in porcine ciliary epithelium via MT3 receptors [49]. This suggests that melatonin may regulate IOP by influencing ion homeostasis.

Additionally, melatonin affects carbonic anhydrase activity, which is crucial for aqueous humor production. Crooke et al. found that 5-MCA-NAT downregulated carbonic anhydrase II (CAII) and XII (CAXII) in rabbit ciliary epithelial cells, leading to an IOP reduction of 51.3%, with effects lasting up to 96 h [48]. These findings suggest that melatonin analogs may provide a prolonged ocular hypotensive effect by modulating carbonic anhydrase activity.

However, several human studies have investigated melatonin’s effects on IOP, yielding mixed results. Viggiano et al. conducted a placebo-controlled study in healthy volunteers and found that oral melatonin had no significant effect on aqueous humor flow, suggesting that systemic melatonin alone does not regulate IOP [50]. While Samples et al. reported that oral melatonin lowered IOP in humans, and this effect was diminished when melatonin synthesis was suppressed by bright light exposure [51]. This suggests that endogenous melatonin levels may contribute to IOP regulation.

Recent clinical trials have explored the therapeutic potential of melatonin analogs for glaucoma treatment. Pescosolido et al. reported that oral agomelatine (a melatonin receptor agonist) reduced IOP by approximately 30% in glaucoma patients who were unresponsive to conventional therapy [52]. Additionally, Ismail and Mowafi found that oral melatonin (10 mg) administered before cataract surgery significantly lowered IOP, reduced anxiety, and improved surgical conditions [53]. These findings indicate that melatonin and its analogs may have clinical applications beyond glaucoma management.

To enhance the efficacy of melatoninergic compounds, researchers have developed innovative drug formulations. Dal Monte et al. tested a nanomicellar formulation of melatonin and agomelatine in a hypertensive glaucoma rat model and found that it significantly prolonged the IOP-lowering effect compared to saline formulations [54]. Another study by Dal Monte et al. demonstrated that a topical melatonin/agomelatine formulation reduced IOP by 60% and preserved retinal ganglion cell function, outperforming standard treatments like timolol and brimonidine [55]. These findings highlight the potential of nanotechnology-based melatoninergic therapies for sustained IOP control.

### 3.2. Neuroprotective Mechanisms of Melatonin in the Retina

Beyond its well-established roles in sleep regulation and IOP reduction, melatonin acts as a potent endogenous antioxidant, contributing critically to redox homeostasis and cytoprotection (Figure 3). Melatonin possesses the remarkable capacity to scavenge up to 10 reactive oxygen and nitrogen species (ROS/RNS) through its metabolic cascade, substantially outperforming classical antioxidants that typically target only a limited range of radicals [56,57,58]. This extensive antioxidative capacity supports melatonin’s protective role in maintaining cellular and tissue oxidative balance.

Melatonin functions in both aqueous and lipid compartments of the cell [59], where it safeguards essential biomolecules—including lipids [60], proteins [61,62,63], and DNA [64,65]—from oxidative damage. These effects are mediated through multiple mechanisms such as radical adduct formation, hydrogen transfer, and single-electron transfer [66]. Furthermore, melatonin’s metabolites retain antioxidant activity, creating a cascade effect that extends its detoxification capacity. Importantly, melatonin effectively neutralizes ROS generated by the electron transport chain (ETC, complexes I–IV) [67,68]. This antioxidative action not only reduces oxidative stress but also enhances ETC efficiency, thereby promoting ATP production [69,70,71,72]. These combined effects underscore melatonin’s critical role in maintaining mitochondrial integrity and optimizing cellular energy metabolism.

In mitochondria, melatonin exerts regulatory effects on membrane dynamics and integrity. It stabilizes the mitochondrial membrane potential via modulation of uncoupling protein 2 (UCP2), inhibits mitochondrial permeability transition pore (MPTP) opening [73,74], and reduces lipid peroxidation to preserve membrane fluidity [60]. These mechanisms collectively limit cytochrome c release under oxidative stress—an event implicated in both glaucoma pathology and neurodegenerative diseases such as Alzheimer’s, Parkinson’s, and Huntington’s diseases, where melatonin also demonstrates neuroprotective efficacy [75,76,77,78,79].

Beyond direct ROS scavenging, melatonin enhances endogenous antioxidant defenses. It upregulates key antioxidant enzymes, including glutathione peroxidase (GPx), superoxide dismutases (SODs), and catalase [80,81], while promoting glutathione (GSH) synthesis via activation of *γ*-glutamylcysteine synthase through AP-1 transcription factor induction [82]. Under oxidative stress, melatonin’s ability to maintain GSH levels surpasses that of conventional antioxidants such as vitamins C and E, underscoring its unique role in redox regulation [83].

Melatonin’s neuroprotective actions further extend to promoting mitochondrial integrity in neurogenic niches. Ramírez-Rodríguez et al. reported that melatonin enhances survival of neuronal progenitor cells and immature neurons in the dentate gyrus of C57BL/6 mice [84], potentially through circadian regulation and melatonin receptor activation. This axis linking mitochondrial function with neurogenesis is of particular relevance to glaucoma-associated neurodegeneration.

At the molecular level, melatonin exhibits anti-apoptotic properties through multiple pathways. It modulates Bcl-2 family proteins [85] to inhibit mitochondrial apoptotic signaling [86,87,88,89]. Radogna et al. demonstrated that melatonin regulates the Bcl-2/Bax ratio via both MT receptor-dependent and -independent mechanisms, thereby preventing cytochrome c release and subsequent caspase activation in U937 cells [90,91,92]. In parallel, melatonin suppresses pro-apoptotic MAPK pathways (JNK/p38) via the Trx/ASK1 axis [93,94,95] and enhances SIRT1 expression [96,97,98], facilitating deacetylation of stress-responsive substrates [99,100,101,102]. Moreover, melatonin modulates autophagy through the PI3K/AKT/mTOR pathway [103,104] and inhibits ferroptosis by activating the Nrf2/Slc7a11 axis to limit lipid hydroperoxide accumulation [105].

Melatonin also exerts robust anti-inflammatory effects by downregulating pro-inflammatory mediators. It inhibits NF-*κ*B-driven transcription of NOS and COX-2 [106,107,108,109], and reduces the expression of cytokines such as IL-1*β*, IL-6, and TNF-*α* [110,111,112]. In optic neuritis models, melatonin has been shown to reduce microglial activation, astrocytosis, demyelination, and retinal ganglion cell loss. It concurrently decreases nitric oxide synthase 2, COX-2, and TNF-*α* levels, while mitigating lipid peroxidation [113]. Liu et al. further demonstrated that melatonin inhibits TNF-*α*, IL-8, and VEGF expression via MT1 receptor-mediated PI3K/Akt and ERK signaling pathways in synovial fibroblasts [114]. Additionally, melatonin suppresses inflammasome activation and pyroptosis by inhibiting the EGR1/DDX3X/NLRP3 axis [115].

## 4. Melatonin Modulates Multi-Faceted Mechanisms Underlying Glaucomatous Neurodegeneration

While elevated IOP remains the primary risk factor for glaucoma, contributing to optic nerve damage through disrupted axonal transport and impaired neuronal signaling, growing evidence indicates that RGC degeneration involves a complex interplay of additional pathological mechanisms. These include vascular dysregulation-induced hypoxia [116], glutamate excitotoxicity [117,118], neuroinflammation [119], oxidative stress [120,121,122], and mitochondrial dysfunction [123,124,125,126,127], all of which may act independently of or synergistically with IOP elevation to accelerate disease progression. Melatonin, a pleiotropic molecule with neuroprotective properties, has shown potential in modulating these processes, offering a multifaceted therapeutic avenue for glaucomatous neurodegeneration [128].

### 4.1. Mitochondrial Dysfunction and Oxidative Imbalance

Mitochondrial dysfunction is a well-established contributor to glaucomatous damage, marked by compromised electron transport chain activity, reduced ATP production, and excessive accumulation of ROS in both RGCs and cells of the outflow pathway (e.g., trabecular meshwork) [129,130]. In primary open-angle glaucoma patients, mitochondrial complex I activity is reduced by approximately 18% in lymphoblasts compared to controls [131,132], further aggravated by mitochondrial DNA mutations and increased membrane permeability [133,134]. These impairments compromise cellular energy metabolism and increase vulnerability to oxidative stress and excitotoxic injury. Importantly, endogenous melatonin levels—rising during adolescence and declining with age—play a crucial role in maintaining mitochondrial homeostasis and redox balance [135,136]. Exogenous melatonin not only scavenges ROS and enhances ATP production but also promotes mitophagy and mitochondrial biogenesis, thereby preserving mitochondrial quality control in stressed RGCs [137].

### 4.2. Glutamate Excitotoxicity and Calcium Dysregulation

Glaucomatous stressors such as elevated IOP or retinal ischemia induce pathological glutamate accumulation, leading to overstimulation of NMDA receptors and subsequent intracellular calcium overload. This triggers mitochondrial calcium dysregulation, activation of caspases, and ultimately RGC apoptosis [138,139,140]. Melatonin protects against excitotoxicity by (1) upregulating glutamate transporters (e.g., GLT-1) through Nrf2-mediated transcription and PI3K/Akt signaling [112]; (2) directly inhibiting NMDA receptor-mediated currents in retinal neurons, reducing calcium overload [141,142]; and (3) enhancing SIRT1/PGC-1*α*-dependent mitochondrial calcium handling, preventing bioenergetic collapse [143,144].

### 4.3. Neuroinflammation and Autophagy Dysregulation

Chronic neuroinflammation, driven by activated microglia and astrocytes, promotes glaucomatous neurodegeneration through the release of pro-inflammatory cytokines such as TNF-*α*, IL-6, and nitric oxide [145,146]. Autophagy, a lysosome-dependent degradation process, plays a dual role—serving as a protective mechanism under physiological conditions but becoming deleterious when dysregulated. In glaucomatous trabecular meshwork cells, oxidative stress impairs lysosomal function, leading to autophagosome accumulation and blocked autophagic flux [147,148]. Melatonin modulates autophagy in a context-dependent manner via the AMPK/mTOR signaling axis: it activates protective autophagy during energy deprivation by inhibiting mTORC1 and activating ULK1, while also preventing excessive autophagic cell death by stabilizing lysosomal integrity through mTORC2 signaling [149,150,151]. This balanced regulation supports cellular survival under stress conditions.

### 4.4. Pyroptosis and Inflammatory Cascades

Pyroptosis, a form of programmed necrosis associated with inflammation, has been increasingly recognized in acute glaucomatous injury. Activation of the CASP8-HIF-1*α* axis leads to cleavage of gasdermin D (GSDMD), which promotes NLRP3 inflammasome activation, caspase-1-mediated IL-1*β* maturation, and amplified inflammatory responses [152]. In experimental high-IOP models, melatonin suppresses pyroptotic signaling by downregulating the NF-*κ*B/NLRP3 pathway, thereby reducing IL-1*β* secretion and preserving retinal structure. This anti-pyroptotic effect complements melatonin’s broader anti-inflammatory actions, including microglial deactivation and cytokine suppression [153]. Furthermore, melatonin’s anti-inflammatory actions extend to glial cells: it limits microglial overactivation and inflammatory cytokine release, thereby creating a more neuroprotective milieu.

## 5. Melatonin Exerts Multi-Target Neuroprotective Effects in Glaucoma Pathogenesis

Accumulating evidence supports melatonin’s multi-target neuroprotective role in glaucoma, acting through antioxidative, anti-apoptotic, anti-inflammatory, and synaptic regulatory mechanisms. These diverse effects are mediated via mitochondrial preservation, inhibition of regulated cell death pathways, immune modulation, and retinal circuit stabilization as listed in Appendix A.

In retinal ischemia–reperfusion (RIR) injury models, Zhang et al. [154] demonstrated that melatonin administration in C57BL/6J mice significantly reduced RGC death (37.2% increase in survival), lipid peroxidation (54% decrease in malondialdehyde levels), and iron accumulation (42% reduction in ferritin expression). Mechanistically, melatonin preserved mitochondrial ultrastructure (28% increase in cristae density) by modulating the Slc7a11/Alox12 axis and inhibiting p53-mediated ferroptosis, thereby mitigating oxidative and iron-dependent cell death.

Its anti-apoptotic efficacy was further highlighted in acute ocular hypertension (AOH) models. Ye et al. [155] reported that melatonin inhibited PANoptosis—a hybrid pathway integrating apoptosis, necroptosis, and pyroptosis—by suppressing caspase-3 (62% reduction in TUNEL-positive cells), RIP1/RIP3 phosphorylation (73% decrease in p-RIP1), and NLRP3 inflammasome activation (58% reduction in IL-1*β*). These effects preserved retinal laminar integrity, with inner nuclear layer thickness retained at 89.7% (*p* < 0.001 vs. controls). Similarly, Shi et al. [156] identified an SIRT1-dependent mechanism in optic nerve crush models, where melatonin reduced p53 acetylation (65% reduction in Ac-p53), elevated antioxidant capacity (1.8-fold increase in SOD activity), and suppressed senescence (42% reduction in SA-*β*-gal-positive cells). The reversal of these effects by EX-527, a SIRT1 inhibitor, confirmed pathway specificity.

In hypertensive glaucoma models, Dal Monte et al. [55] showed that topical melatonin/agomelatine application reduced IOP by 28.4% and inhibited Bax-caspase-3-mediated apoptosis (61% reduction in caspase-3 activity). However, in contrast, Marangoz et al. [157] found that systemic melatonin administration in episcleral vein-cauterized rats lowered IOP by 19.2%, but failed to significantly preserve RGC density—unlike brimonidine tartrate, which maintained 78.4% RGC survival. These contrasting results suggest that melatonin’s neuroprotective efficacy may be influenced by the delivery route, bioavailability, and disease stage, highlighting the need for pharmacokinetic optimization in clinical applications.

Beyond acute injury models, melatonin also exhibits benefits in chronic, normotensive glaucoma contexts. Hu et al. [158] showed that oral melatonin improved RGC survival (to 82.3% of control levels) in EAAC1−/− mice via the NRF2/p53/SIRT1 axis, with concurrent reductions in oxidative DNA damage (54% decrease in 8-OHdG-positive cells). In addition, González Fleitas et al. [159] reported that melatonin selectively protected melanopsin-expressing RGCs (76% survival increase) and restored mitochondrial membrane potential (2.3-fold increase in JC-1 red/green fluorescence ratio), leading to functional recovery in circadian light reflexes (41% decrease in latency).

Melatonin’s neurorestorative effects extend to synaptic function. Zhao et al. [160] demonstrated that MT2 receptor activation enhanced glycine receptor currents by 217% in rat RGCs via a phosphatidylcholine-specific PLC/PKC pathway, independent of cAMP/cGMP signaling. This modulation improved contrast sensitivity under scotopic conditions (peak efficacy at 5 lux), suggesting its relevance in visual function preservation.

The immunomodulatory actions of melatonin were elucidated in NMDA-induced excitotoxicity models. Zou et al. [161] showed that melatonin suppressed microglial activation (63% reduction in Iba1+ area), decreased TNF-*α* secretion (58%), and inhibited p38 MAPK signaling in RGCs (72% decline in p-p38). These effects were abolished by minocycline, confirming a microglia-dependent mechanism. In complementary work, Belforte et al. [162] found that melatonin restored glutamate-GABA balance (38% reduction in glutamate; 1.5-fold increase in GAD67 expression) and attenuated nitric oxide overproduction (64% reduction in iNOS), thereby limiting neuroinflammatory cascades. These findings highlight the translational need for optimized delivery and receptor-targeted design.

## 6. Melatonin Delivery Strategies for Ocular Application

Current clinical studies on melatonin for glaucoma therapy primarily employ oral administration. However, its clinical utility remains limited due to the lack of robust preclinical models and its inherently low oral bioavailability (15%), which is attributed to extensive hepatic first-pass metabolism [163]. Topical ocular delivery, though widely considered, faces significant physiological barriers such as rapid tear turnover, corneal impermeability, nasolacrimal drainage, and enzymatic degradation. These factors contribute to poor drug retention, low targeting efficiency, and limited transcorneal permeation [164]. Alternatively, while intravitreal injections can deliver drugs directly to the posterior segment, they pose risks of retinal toxicity at high doses [165]. Immediate-release formulations further compound these limitations by exhibiting short plasma half-lives and inadequate absorption profiles [166].

To overcome these challenges, nanotechnology-based drug delivery systems have been increasingly explored. Nanocarriers with mucoadhesive and mucopenetrating properties—such as nanocapsules, liposomes, micelles, and polymeric nanoparticles—show promise in enhancing melatonin permeability across ocular barriers, prolonging retention time, and enabling controlled release [55,167,168,169,170,171,172,173]. For instance, Romeo et al. developed melatonin-loaded lipid–polymer hybrid PLGA-PEG nanoparticles coated with a cationic lipid shell, which demonstrated superior neuroprotective and antioxidant activity compared to melatonin in aqueous solution [174]. Similarly, Quinteros et al. reported that combining bioadhesive polymers with liposomal carriers significantly improved the IOP-lowering efficacy of the melatonin derivative 5-MCA-NAT [167].

Nanocarrier systems can be categorized by geometry: (1) Zero-dimensional (0D) nanoparticles—Polymeric nanoparticles, such as poly(lactic-*co*-glycolic acid)-polyethylene glycol (PLGA-PEG), have been shown to sustain melatonin release for up to 8 h and produce greater IOP reduction than solution-based formulations [172,175]. Cationic nanoparticles, such as those coated with chitosan, enhance mucoadhesion and prolong corneal residence time [169]. (2) One-dimensional (1D) nanofibers—Electrospun nanofibers made from poly(vinyl alcohol) (PVA) and poly(lactic acid) (PLA) exhibit tunable release profiles: PVA achieves rapid, complete drug release within 20 min, while PLA supports sustained delivery over several hours, thus improving ocular bioavailability [173,176]. (3) Two-/three-dimensional (2D/3D) systems—Nanofilms and hydrogels, particularly those based on hyaluronic acid, offer high drug-loading capacity and stimulus-responsive release. However, their application in melatonin delivery remains underexplored [177].

Therefore, tailoring melatonin delivery methods—such as optimized dosing schedules, patient-specific formulation selection (e.g., nanocarriers, topical application, or sustained-release systems), and precise receptor targeting—could enhance both safety and effectiveness. These approaches are particularly relevant in overcoming current limitations related to bioavailability and therapeutic consistency.

Despite the promising advances in nanocarrier-based systems, several obstacles remain. These include limitations in penetration efficiency, therapeutic consistency, and ocular targeting. Future research should focus on optimizing delivery vectors, improving tissue-specific transport, and validating efficacy in clinically relevant models. Furthermore, successful clinical translation will require systematic evaluation of safety, therapeutic effectiveness, scalability, commercial viability, and adherence to regulatory standards.

## 7. Discussion

Melatonin represents a promising yet underutilized candidate in the therapeutic landscape of glaucoma. Its dual capacity to modulate intraocular pressure and exert neuroprotective effects offers a pharmacological advantage over agents that solely target pressure reduction. However, despite encouraging preclinical evidence, the clinical application of melatonin remains limited, and the heterogeneity in study design, formulation types, and dosing regimens poses significant barriers to translation. As this review has shown, most investigations are restricted to animal models or small-scale human trials, thereby limiting generalizability.

One major limitation encountered during this review was the restricted number of high-quality, peer-reviewed clinical studies that specifically address melatonin-based interventions for glaucoma. Although melatonin and its analogs have been extensively studied in the context of neurodegeneration, their targeted role in ocular pathophysiology remains comparatively less explored. Despite efforts to broaden the search strategy, the lack of standardized endpoints and inconsistent reporting of outcomes hindered direct comparisons between studies. Additionally, the current literature does not adequately address the long-term safety profile of melatonin, particularly when administered in sustained-release or high-dose formulations tailored for ocular delivery.

Emerging human pluripotent stem cell-derived retinal organoids can recapitulate key features of glaucomatous injury—such as RGC axonal degeneration, gliosis, and neuroinflammation—when subjected to controlled hydrostatic pressure (e.g., 15–30 mmHg). We propose that future work leverage these organoid models to assess the neuroprotective and IOP-modulating effects of melatonin and its analogs. Such studies should measure RGC survival, neurite integrity, synaptic marker expression, and inflammatory cytokine release within perfused organoid culture systems. Comparative evaluation of oral, topical, and nanocarrier-based delivery in organoid perfusion will help define optimal dosing regimens for clinical translation [178].

Given the complexity of glaucomatous neurodegeneration, future research should prioritize well-designed clinical trials to determine optimal delivery strategies, dose safety, and patient-specific responsiveness to melatonin-based therapies. Moreover, elucidating the interplay between melatonin receptors, circadian rhythm disruption, and retinal ganglion cell survival may uncover new therapeutic windows. This review underscores the need for multidisciplinary efforts to validate melatonin’s translational potential in glaucoma management.

## 8. Conclusions

Glaucoma is a complex neurodegenerative disorder in which retinal ganglion cell loss and optic nerve damage result from not only elevated IOP, but also mitochondrial dysfunction, oxidative stress, neuroinflammation, and excitotoxicity. Melatonin, through its pleiotropic actions—including antioxidant, anti-inflammatory, anti-apoptotic, and mitochondrial protective effects—has shown significant neuroprotective potential in experimental glaucoma models. Its ability to modulate multiple pathogenic pathways positions it as a compelling candidate for disease-modifying therapy.

Melatonin regulates IOP via MT1, MT2, and MT3 receptor pathways and demonstrates additive or synergistic effects when combined with conventional ocular hypotensives. However, its systemic bioavailability remains a major barrier to clinical translation. Recent advances in nanocarrier-based delivery systems—such as polymeric nanoparticles, nanofibers, and hydrogels—offer promising solutions for enhancing ocular retention, permeability, and sustained release.

To fully harness the therapeutic potential of melatonin in glaucoma, future efforts should focus on optimizing delivery vectors, validating receptor-specific mechanisms, and conducting well-designed clinical trials. With continued development, melatonin-based therapeutics may offer a novel, multi-target strategy for preserving visual function in glaucoma patients.

## Figures and Tables

**Figure 1 biomedicines-13-01213-f001:**
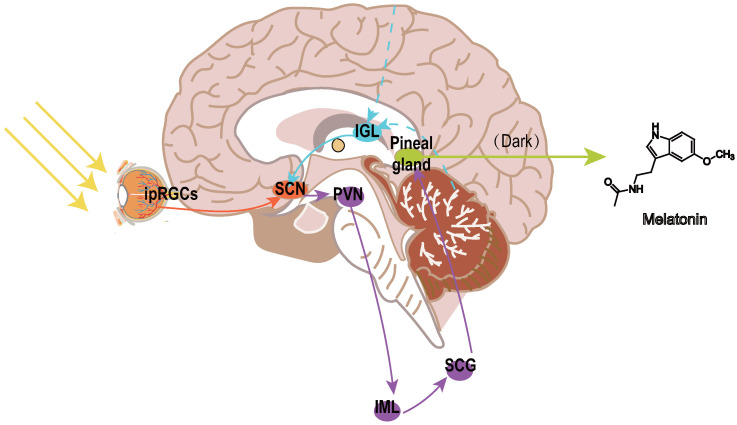
Neural pathways regulating circadian rhythms and melatonin secretion. The retinohypothalamic tract (RHT; orange arrows) originates from melanopsin-expressing intrinsically photosensitive retinal ganglion cells (ipRGCs) in the retina, transmitting light signals to the suprachiasmatic nucleus (SCN) via glutamate and pituitary adenylate cyclase-activating polypeptide (PACAP). The geniculohypothalamic tract (GHT; blue arrows) originates from the intergeniculate leaflet (IGL) of the thalamus, integrating non-photic behavioral cues (e.g., locomotion, social interaction) through neuropeptide Y (NPY) and GABAergic inputs to modulate SCN activity. The SCN projects to the pineal gland via a multisynaptic pathway: SCN → paraventricular nucleus (PVN) → spinal intermediolateral column (IML) → superior cervical ganglion (SCG) → pineal gland. Melatonin (purple gradient) is rhythmically released in darkness and suppressed by light. Abbreviations: SCN, suprachiasmatic nucleus; IGL, intergeniculate leaflet; PVN, paraventricular nucleus; IML, intermediolateral column; SCG, superior cervical ganglion; RHT, retinohypothalamic tract; GHT, geniculohypothalamic tract; ipRGCs, intrinsically photosensitive retinal ganglion cells.

**Figure 2 biomedicines-13-01213-f002:**
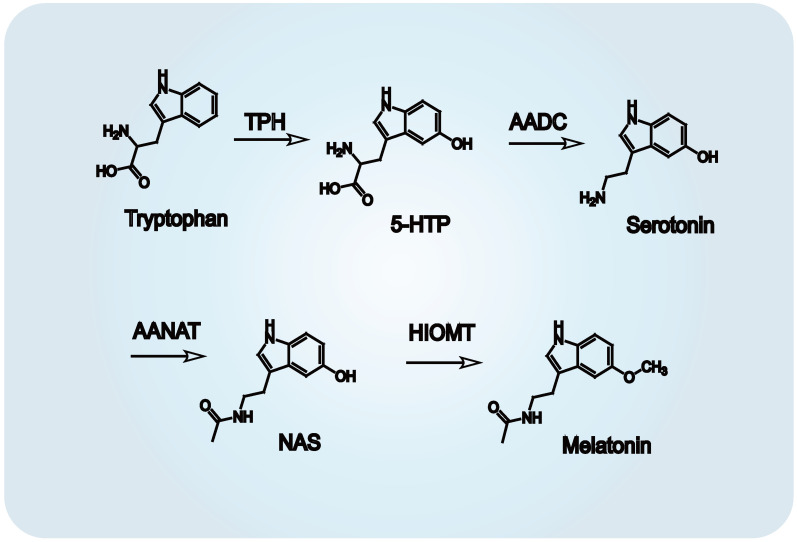
Biosynthetic pathway of melatonin from tryptophan. Melatonin synthesis occurs via four enzymatic steps: (1) Tryptophan hydroxylase (TPH) catalyzes hydroxylation of tryptophan to 5-hydroxytryptophan. (2) Aromatic amino acid decarboxylase (AADC) decarboxylates 5-hydroxytryptophan to serotonin (5-hydroxytryptamine). (3) Arylalkylamine *N*-acetyltransferase (AANAT) acetylates serotonin to *N*-acetylserotonin. (4) Hydroxyindole-*O*-methyltransferase (HIOMT) transfers a methyl group to *N*-acetylserotonin, yielding melatonin *N*-acetyl-5-methoxytryptamine).

**Figure 3 biomedicines-13-01213-f003:**
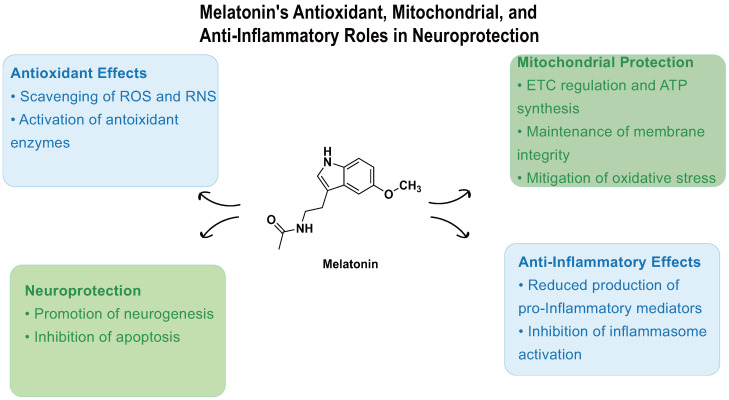
Melatonin-mediated neuroprotective mechanisms. Melatonin acts through four major pathways: (1) antioxidant defense —scavenging ROS/RNS and upregulating GPx, SOD, catalase, and GSH synthesis; (2) mitochondrial protection—enhancing ETC efficiency, stabilizing membrane potential via UCP2, inhibiting MPTP opening, and preventing cytochrome c release; (3) anti-apoptotic regulation—modulating Bcl-2/Bax balance and suppressing MAPK signaling pathways; and (4) anti-inflammatory response—downregulating NF-*κ*B, IL-1*β*, IL-6, and TNF-*α*, and inhibiting the NLRP3 inflammasome via the EGR1/DDX3X axis. These mechanisms collectively support retinal ganglion cell survival and reduce glaucomatous neurodegeneration.

## Data Availability

Data sharing is not applicable.

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
