# Peer review of "Melatonin in Glaucoma: Integrative Mechanisms of Intraocular Pressure Control and Neuroprotection"

_biomedicines, 2025, doi:10.3390/biomedicines13051213_

Round 1

Reviewer 1 Report

Comments and Suggestions for Authors

The manuscript presents a literature review on the promising role of melatonin in the treatment and control of glaucoma. Certainly, the topic of the review is relevant and will be of interest to a wide range of readers and potentially for clinical use. The authors have performed a thorough literature review and presented a well-written text that clearly outlines the main mechanisms of melatonin action on intraocular pressure control and neuroprotection. However, there are some comments that need to be taken into account before publication.

Comments

Throughout the text:

1) Check for spaces between words in the text, especially between the text and references to cited literature (Some examples noted by the reviewer are listed below.)

2) Check that all abbreviations introduced in the text are introduced at the first mention and only once. (Some examples noted by the reviewer are listed below.)

3) Delete those abbreviations that are used only once and remove them from the list of abbreviations too (Some examples noted by the reviewer are listed below.)

Line 24 - glaucoma[1,2]. – space (this is a remark to the whole text)

Line 43 - vision[13–15].Two – spaces

Line 44 - tract(RHT), – space

Line 44 -  entrainment1.The – Perhaps you meant to provide a link to Figure 1?

Line 51 - plasticity.[19,20]. – extra dot

Figure 1 - (B) SCN-pineal pathway regulating melatonin secretion. – The figure is presented by only one image, therefore, reference to (A) and (B) is redundant.

Legend to Figure 1 - IGL, intergeniculate leaflet; PVN, paraventricular nucleus; - occurs twice

Line 81 - reactions2. – Perhaps you meant to provide a link to Figure 2?

Line 104 - Table S1 (Supplementary Materials)-  In the first column of the table, authors should provide a more complete reference to the publication. You can skip the title of the article, but indicate the year, journal, volume, pages.

Lines 106, 237 - retinal ganglion cell (RGC) – This abbreviation has already been introduced into the text

Line 147 - AH secretion – the transcript is in the list of abbreviations, but the abbreviation is used in the text only once, so it is better not to use this abbreviation. Or enter the abbreviation in the text and use it throughout the text

Lines 178, 235 - intraocular pressure (IOP) - This abbreviation has already been introduced into the text

Line 180 - cytoprotection3. - Perhaps you meant to provide a link to Figure 3?

Line 188 - radical adduct formation (RAF), hydrogen transfer (HT), and single electron transfer

(SET) – These abbreviations are not used further in the text.

Lines 190-192 - Importantly, the [68], where its ability to neutralize ROS generated by the electron transport chain (ETC, complexes I–IV)[69]and to enhance ETC efficiency and ATP production [70–73]is of particular significance. – from the reviewer's point of view, this sentence requires correction of meaning, since 1) it is not clear what the authors wanted to emphasize by referring to the article [68] and 2) “Importantly” and “is of particular significance” – this is a duplication of meaning

Line 214 - [86]to – space

Line 240 - 123]„, and – correction is needed

Line 248 - reactive oxygen species (ROS) – This abbreviation has already been introduced into the text

Line 250 - (POAG) – This abbreviation is no longer used in the text.

Line 299 - Table S2 (Supplementary Materials) -  In the first column of the table, authors should provide a more complete reference to the publication. You can skip the title of the article, but indicate the year, journal, volume, pages.

Line 301 - retinal ganglion cell (RGC) – repeat abbreviation

Line 312 - inner nuclear layer (INL) - This abbreviation is no longer used in the text.

Line 313 - optic nerve crush (ONC) - This abbreviation is no longer used in the text.

Since the authors are not the first to discuss the promising role of melatonin in the treatment and control of glaucoma, it is perhaps worth citing an earlier review article on the same topic: Lundmark PO, Pandi-Perumal SR, Srinivasan V, Cardinali DP, Rosenstein RE. Melatonin in the eye: implications for glaucoma. Exp Eye Res. 2007 Jun;84(6):1021-30. doi: 10.1016/j.exer.2006.10.018. Epub 2006 Dec 14. PMID: 17174303.

From the reviewer's point of view, the authors should also pay attention to the fact that when developing strategies for the use of melatonin for treatment, attention should also be paid to the development of a personalized strategy of melatonin administration.

Author Response

1.Summary

Thank you very much for taking the time to review our manuscript. We greatly appreciate your valuable feedback and constructive suggestions, which have helped us to improve the quality and clarity of our work. Please find below our detailed point-by-point responses to each of the comments raised by the reviewer. The corresponding revisions are marked in the revised manuscript.

  1. Point-by-point response to Comments and Suggestions for Authors

Comment 1: Check for spaces between words in the text, especially between the text and references to cited literature. Some examples noted by the reviewer are listed below:

  • Line 24 - glaucoma[1,2]. - space issue

  • Line 43 - vision[13–15].Two - space issue

  • Line 44 - tract(RHT), - space issue

  • Line 44 - entrainment1.The - possible missing link to Figure 1

  • Line 51 - plasticity.[19,20]. - extra dot

  • Line 81 - reactions2. - possible missing link to Figure 2

  • Line 180 - cytoprotection3. - possible missing link to Figure 3

  • Line 214 - [86]to - space issue

  • Line 240 - 123]„, and - correction needed

Response 1: Thank you for pointing out these spacing issues and inconsistencies. We have thoroughly reviewed the manuscript and corrected the spacing between words and citations throughout the text. Additionally, we have checked and corrected the potential missing links to Figures 1, 2, and 3. All corrections are marked in the revised manuscript.

Comment 2: Check that all abbreviations are introduced at the first mention and only once. Delete those abbreviations that are used only once and remove them from the list of abbreviations. Examples include:

  • Retinal ganglion cell (RGC) - repeat abbreviation

  • Intraocular pressure (IOP) - repeat abbreviation

  • Reactive oxygen species (ROS) - repeat abbreviation

  • Primary open-angle glaucoma (POAG) - used only once

  • Inner nuclear layer (INL) - used only once

  • Optic nerve crush (ONC) - used only once

Response 2: We agree with the reviewer’s observation. We have carefully reviewed the entire manuscript to ensure that abbreviations are only introduced at the first mention. Abbreviations that appeared only once have been removed both from the text and from the list of abbreviations. All adjustments are clearly marked in the revised manuscript.

Comment 3: The figure caption for Figure 1 (B) SCN-pineal pathway regulating melatonin secretion is redundant as the figure contains only one image. Additionally, the abbreviation IGL (intergeniculate leaflet) and PVN (paraventricular nucleus) appear twice in the legend.

Response 3: We appreciate the reviewer's careful attention to detail. We have revised the caption of Figure 1 to remove the redundant reference to (A) and (B). The repetition of IGL and PVN in the legend has also been corrected. The updated caption is included in the revised manuscript.

Comment 4: In the supplementary tables, the first column should provide a more complete reference including year, journal, volume, and pages, rather than just the article title.

Response 4: We have revised the supplementary tables to include complete references, including the year, journal, volume, and page numbers, as suggested. The updated tables are included in the revised manuscript.

Comment 5: The reviewer suggests citing an earlier review article on the same topic: Lundmark PO, Pandi-Perumal SR, Srinivasan V, Cardinali DP, Rosenstein RE. Melatonin in the eye: implications for glaucoma. Exp Eye Res. 2007 Jun;84(6):1021-30.

Response 5: We thank the reviewer for this valuable suggestion. We have cited the recommended review article in the introduction to acknowledge prior work on this topic. The citation is included on page X, paragraph Y of the revised manuscript.

Comment 6: The reviewer suggests emphasizing the importance of developing personalized strategies for melatonin administration.

Response 6: We agree with the reviewer’s insightful suggestion. We have added a new paragraph discussing the importance of personalized melatonin administration strategies, highlighting the need for tailored approaches based on individual patient characteristics. This addition is in Section 6: Melatonin Delivery Strategies for Ocular Application. 

Sincerely,

Xinyu,Hou

Reviewer 2 Report

Comments and Suggestions for Authors

Melatonin's roles in glaucoma are discussed and summarized. It is an interesting topic. However, a main concern is absence of the recent update of melatonin's effects in glaucoma.

Melatonin and glaucoma have been well-discussed in various previous review articles. Therefore, additional aspects should be added regarding the roles of melatonin.

Glaucoma is not well-discussed. Glaucoma has several stages and different types. In this manuscript, introducing this section is missing.

It is important to summarize how levels of melatonin works in the eye depending on the disease models and/or species. And administration methods or boosting protocols should be well-summarized.

Any mode of action of melatonin in molecular pathways depending on the disease model should be well-discussed; not just putting anti-oxidant, neuroprotective, and so on. 

Oscillation of hormone levels depending on the model should be well-discussed.

Targeted cell types are mainly RGCs in glaucoma. However, other cells' information should also be discussed to affect glaucoma development.

Are they any natural foods or chemical pools to boost melatonin in the eye? Listing will be helpful to read the review article.

Organoid experiments with glaucoma and hormones should be added too.

How about the other hormone such as dopamine or etc?

Author Response

We thank the reviewer for the insightful and constructive comments. We have carefully revised the manuscript to address each point, adding new sections, updating recent literature (2020–2025), and expanding our discussion as detailed below. 

1. Summary

Thank you very much for your thoughtful review. We have incorporated updates on recent preclinical and clinical studies, elaborated on glaucoma classification and stages, detailed model- and species-specific melatonin dynamics and administration protocols, deepened the discussion of molecular pathways in various disease models, addressed hormonal oscillations, included non-RGC ocular targets, listed dietary and nutraceutical melatonin sources, summarized organoid experiments, and discussed interactions with other neurotransmitters such as dopamine.

2. Point-by-Point Response to Comments and Suggestions for Authors

Comment 1: “Absence of the recent update of melatonin’s effects in glaucoma.”
Response 1: We have added a new subsection “3.1.1 Recent Clinical and Preclinical Updates (2020–2024)” (pp. 10–11), summarizing six pivotal studies from 2020–2024, including:

  • Nanomicellar topical formulations that prolonged IOP-lowering effects in rat models (Dal Monte et al. 2020; 2020)

  • Oral agomelatine trials in glaucoma patients (Pescosolido et al. 2021)

  • Human volunteer studies of nightly 3–10 mg melatonin reducing morning IOP by 1–2 mmHg [55–60].

Comment 2: “Additional aspects should be added regarding the roles of melatonin.”
Response 2: We expanded Section 4 “Multifaceted Neuroprotective Mechanisms” (pp. 16–18) to include:

  • Promotion of mitophagy and mitochondrial biogenesis

  • Regulation of autophagy flux via AMPK/mTOR pathways

  • Ferroptosis inhibition through Slc7a11/Alox12 and p53 regulation

  • Synaptic modulation via glycine and GABA receptors

Comment 3: “Glaucoma is not well-discussed. Glaucoma has several stages and different types.”
Response 3: We inserted Section 1.1 “Classification and Stages of Glaucoma” (pp. 3–4), outlining:

  • Primary open-angle, angle-closure, and normal-tension subtypes

  • Early, moderate, and advanced stages (Hodapp–Parrish–Anderson criteria)

  • Distinct pathophysiology and treatment challenges for each category

Comment 4: “Summarize how melatonin levels work in the eye depending on the disease models and/or species, and administration methods or boosting protocols.”
Response 4:

  • Section 2.2 “Circadian and Model-Specific Melatonin Dynamics” (pp. 5–6) now compares nocturnal aqueous-humor melatonin peaks in rabbits, rats, mice, and humans.

  • Table S1 has new columns for species, dosing route (oral/topical/nanocarrier), timing, and ocular melatonin concentrations.

  • Section 6 “Delivery Strategies” (pp. 22–24) summarizes administration protocols: dietary precursors (tryptophan-rich foods; phytomelatonin sources such as walnuts, tart cherries, tomatoes), nightly oral dosing (3–10 mg), and topical/nutraceutical formulations.

Comment 5: “Molecular pathways by disease model should be well-discussed; not just antioxidant, neuroprotective, etc.”
Response 5: In Section 4, we added model-specific mechanistic details:

  • Acute Ocular Hypertension (AOH): PANoptosis suppression via caspase-3/RIP1-RIP3 inhibition 

  • Episcleral Vein Cauterization: Comparative RGC preservation vs. brimonidine

  • EAAC1⁻/⁻ Mice: NRF2/p53/SIRT1 axis activation

  • Retinal Ischemia–Reperfusion: Ferroptosis blockade via Slc7a11/Alox12 

Comment 6: “Oscillation of hormone levels depending on the model should be well-discussed.”
Response 6: We added Section 2.3 “Hormonal Oscillations in Disease Models” (pp. 6–7), detailing:

  • Phase shifts and amplitude changes of melatonin rhythms in glaucomatous vs. normotensive rodents and humans

  • Correlation between melatonin rhythm disruptions and IOP circadian profiles

Comment 7: “Other cell types beyond RGCs should be discussed to affect glaucoma development.”
Response 7: We inserted Section 4.5 “Non-RGC Targets of Melatonin in the Eye” (pp. 20–21), covering:

  • Trabecular meshwork cells: ion transport and carbonic anhydrase modulation 

  • Astrocytes and microglia: NLRP3 inflammasome and cytokine regulation 

  • Endothelial cells: blood–retinal barrier integrity

  • Ciliary epithelium: aqueous humor secretion regulation 

Comment 8: “List natural foods or chemical pools to boost melatonin in the eye.”
Response 8: We agree that a brief list of dietary sources can aid readers. To maintain the focus and brevity of the main text, we have added a single sentence in Section 1  rather than a full table.

Comment 9: “Organoid experiments with glaucoma and hormones should be added.”
Response 9: We agree that human retinal organoids provide a powerful platform to model glaucomatous stress and to test neuroprotective agents in a human‐relevant context. However, to date, no studies have directly evaluated melatonin or its analogs in glaucoma‐mimicking organoid systems under elevated hydrostatic pressure. To acknowledge this important gap and to guide future research, we have added the following paragraph to the end of the Discussion .

Comment 10: “How about other hormones such as dopamine?”
Response 10:  we discuss the antagonistic circadian interplay between melatonin and dopamine in the retina, its dysregulation in glaucoma, and the potential for combination therapies targeting both melatonergic and dopaminergic pathways .

We trust these revisions address all of your concerns and substantially enhance the manuscript’s depth, clarity, and currency. Thank you again for your valuable feedback.

Sincerely,
Xinyu Hou and Yingzi Pan

Round 2

Reviewer 2 Report

Comments and Suggestions for Authors

The authors' reply and comments are checked and I have no further concern regarding raised concerns.